# MicroRNAs: Important Regulatory Molecules in Acute Lung Injury/Acute Respiratory Distress Syndrome

**DOI:** 10.3390/ijms23105545

**Published:** 2022-05-16

**Authors:** Qianying Lu, Sifan Yu, Xiangyan Meng, Mingyu Shi, Siyu Huang, Junfeng Li, Jianfeng Zhang, Yangfan Liang, Mengjun Ji, Yanmei Zhao, Haojun Fan

**Affiliations:** 1Institute of Disaster and Emergency Medicine, Tianjin University, Tianjin 300072, China; qianying.lu@tju.edu.cn (Q.L.); yusifan@tju.edu.cn (S.Y.); mengxiangyan@tju.edu.cn (X.M.); shimingyu1997@tju.edu.cn (M.S.); 2019435008@tju.edu.cn (S.H.); lijunfeng@tju.edu.cn (J.L.); jianfeng_zhang1231@tju.edu.cn (J.Z.); liangyangfan1@163.com (Y.L.); jimengjun2000@163.com (M.J.); 2Tianjin Key Laboratory of Disaster Medicine Technology, Tianjin 300072, China

**Keywords:** MicroRNAs, acute lung injury, acute respiratory distress syndrome, inflammation, biomarker

## Abstract

Acute lung injury (ALI)/acute respiratory distress syndrome (ARDS) is an overactivated inflammatory response caused by direct or indirect injuries that destroy lung parenchymal cells and dramatically reduce lung function. Although some research progress has been made in recent years, the pathogenesis of ALI/ARDS remains unclear due to its heterogeneity and etiology. MicroRNAs (miRNAs), a type of small noncoding RNA, play a vital role in various diseases. In ALI/ARDS, miRNAs can regulate inflammatory and immune responses by targeting specific molecules. Regulation of miRNA expression can reduce damage and promote the recovery of ALI/ARDS. Consequently, miRNAs are considered as potential diagnostic indicators and therapeutic targets of ALI/ARDS. Given that inflammation plays an important role in the pathogenesis of ALI/ARDS, we review the miRNAs involved in the inflammatory process of ALI/ARDS to provide new ideas for the pathogenesis, clinical diagnosis, and treatment of ALI/ARDS.

## 1. Introduction

Acute lung injury (ALI), a life-threatening condition, is caused by direct or indirect injury factors, such as sepsis, pneumonia, inhalation of stomach contents, invasion of epidemic viruses, trauma, blast, smoke inhalation and asphyxiant inhalation [1,2,3,4,5,6,7]. These injury factors activate immune cells, which subsequently initiate abnormal immune responses. Immune cells activated by injury factors release excessive proinflammatory factors, chemokines, and proteases, which ultimately damage lung parenchymal cells, such as vascular endothelial cells and alveolar epithelial cells [2]. When a large number of damaged lung parenchymal cells cause ventilation dysfunction, rapid attenuation of respiratory function occurs [2]. The characteristics of ALI include increased permeability of alveolar epithelial cells and capillary endothelial cells, diffuse pulmonary interstitial and alveolar edema, gas exchange disorders, and hypoxemia, which eventually lead to respiratory failure [8,9]. The clinical emergence of critical illness is mainly characterized by accelerated respiratory frequency, respiratory distress, and intractable hypoxemia. If not treated in time, it will develop into acute respiratory distress syndrome (ARDS) [10,11]. For the last 20 years, the mortality rate of ARDS has remained at approximately 40% [11,12]. In recent years, with the outbreak of acute infectious pneumonia caused by SARS-CoV and SARS-CoV-2, ALI and more serious ARDS have been one of the main causes of death [13,14]. Currently, no effective drug treatment is available to improve the survival of ALI/ARDS patients.

MicroRNAs (miRNAs) are a type of highly conserved single-stranded RNA with a length of approximately 22 nucleotides that are involved in the physiological and pathological functions of a variety of diseases, including tuberculosis [15], ALI/ARDS [16], pulmonary fibrosis [17], hepatitis [18], cardiovascular diseases [18], and cancer [18]. MiRNAs mainly bind to the 3′-untranslated region (3′-UTR) of mRNA to control multiple pathways and various cellular processes, such as inflammatory-immune responses and cell-cell interactions [19]. At present, it is believed that excessively activated and recruited inflammatory cells in the lung produce a large number of proinflammatory factors, and their interaction with effector cells is the main pathophysiological change in ALI/ARDS [2,20,21]. As an important inflammatory regulator, miRNAs might play an important role in ALI/ARDS. Indeed, studies have found that miRNA expression abnormalities can be observed in ALI/ARDS. For example, a clinical trial study found that the expression of miR-150 is low in the serum of ARDS patients and is negatively correlated with the Acute Physiology and Chronic Health Evaluation (APACHE) II score, an indicator to assess the condition and prognosis of ICU patients [22]. Additionally, high miR-122 expression is related to the severity and prognosis of ARDS patients, and miR-122 combined with the APACHE II score has a high evaluation value for the prognosis of ARDS patients [23]. Given that miRNAs play an important role in inflammation, which is a common feature of ALI/ARDS, miRNAs may be diagnostic indicators and therapeutic targets of ALI/ARDS [24]. This article reviews the miRNAs involved in the inflammatory process of ALI/ARDS to provide new ideas for the pathogenesis, clinical diagnosis, and treatment of ALI/ARDS.

## 2. Mechanisms Leading to Tissue Damage in ALI/ARDS

### 2.1. PRRs and Related Molecules in ALI/ARDS

Pattern recognition receptors (PRRs) are an important component of the first line of defense in the natural immune system [25] and play an essential role in the innate and adaptive immune responses of ALI/ARDS. PRRs are expressed by innate immune cells, such as natural killer cells (NK cells), macrophages, neutrophils, and dendritic cells, and mainly include transmembrane pattern recognition receptors (Toll-like receptors (TLRs) and C-type lectin receptors), cytoplasmic pattern recognition receptors (nucleotide-binding oligomerization domain (NOD)-like receptors (NLRs)), retinoic acid-inducible gene I (RIG-I)-like receptors) and cytoplasmic DNA sensors, which can recognize pathogen-associated molecular patterns (PAMPs) and damage-associated molecular patterns (DAMPs) [25]. PAMPs are mainly derived from exogenous molecules of pathogenic or nonpathogenic microorganisms, such as lipopolysaccharide (LPS) in the outer membrane of Gram-negative bacteria, and viral double-stranded RNA (dsRNA) [26]. DAMPs refer to endogenous molecules released after body cells are damaged or die, such as high mobility histone B1 [26] and heat shock protein [26]. These are released after tissues or cells are stimulated via endogenous danger signals, including injury, hypoxia, and stress. Given the numerous reports on TLR and NLR related to ALI/ARDS, the following mainly introduces the structure and pathway of the two families.

#### 2.1.1. TLRs

TLRs are a type of transmembrane protein that play an important role in the innate immune system and the inflammatory process [27]. TLRs are expressed in epithelial cells, dendritic cells, neutrophils, and macrophages and are widely distributed in various parts of the body [28]. TLRs recognize foreign pathogens or cell damage and play a key role in the immune response and the body’s resistance to foreign pathogens.

Studies have found that LPS can activate TLR4 and initiate the expression of downstream inflammatory response signaling pathways, which ultimately result in the release of the inflammatory factors IL-1β, IL-6, and TNF-α [29]. Current studies have confirmed that TLR4 is the most important PRR for recognizing LPS inflammatory signal transduction. Genetic inactivation of TLR4 can significantly attenuate ALI/ARDS [30].

#### 2.1.2. NLRs

NLRs, a type of cytoplasmic pattern recognition receptor, play a unique regulatory role in innate immunity and adaptive immunity. Unlike TLRs that recognize PAMPs on the surface, NLRs are activated by the signals of PAMPs and DAMPs (such as changes in cell volume, rupture of lysosomes, production of reactive oxygen species (ROS), K^+^ efflux, and Ca^2+^ signaling [31]) inside the cell and form part of the inflammasome. Inflammasomes can activate caspase-1 and control the maturation and secretion of IL-1β and IL-18 [32]. The inflammasome mediates the occurrence and development of ALI, so targeted treatment of the inflammasome may represent an effective strategy to prevent and treat ALI.

### 2.2. Inflammation-Related Pathways of ALI/ARDS

#### 2.2.1. TLRs Mediate NF-κB Signaling Pathway

Except for TLR3, all TLRs induce inflammation through the classic signaling of the MyD88-dependent signaling pathway. Myeloid differentiation factor 88 (MyD88) is a key adaptor of the Toll-like receptor/interleuκin-1 receptor (TLR/IL-1 receptor, TIR) pathway [33]. This pathway is closely related to a conserved intracellular sequence of TLRs called TIR [34]. The activated TLR intracellular domain TIR binds to MyD88, and MyD88 binds to interleukin-1 receptor-associated kinase (IRAK), thereby inducing IRAK autophosphorylation. Phosphorylated IRAK1 can bind to and activate tumor necrosis factor receptor-associated factor 6 (TRAF6), which subsequently leads to nuclear factor kappaB (NF-κB) and activator protein-1 (AP-1) activation and eventually induces the expression and release of proinflammatory cytokines, such as IL-1, IL-6, IL-8, TNF-α, and chemokines [35,36] (Figure 1).

Among TLRs, current research has confirmed that TLR4 is the most important PRR that recognizes LPS inflammatory signal transduction. The TLR4-mediated NF-κB signaling pathway is closely related to the occurrence of ALI/ARDS [37]. Studies have shown that MyD88 gene knockout can inhibit NF-κB activation, downregulate the expression and release of TNF-α, and alleviate ALI/ARDS [33]. In addition, TLR4 can also activate the phosphatidylinositol 3-kinase (PI3K) pathway, which is closely related to cell survival, growth, angiogenesis, and metabolism. TLR4 activation by LPS promotes PI3K and its downstream molecule protein kinase B (PKB, also known as AKT) to form a complex with MyD88, which subsequently leads to NF-κB activation [38]. Therefore, controlling the TLR-mediated NF-κB pathway can reduce the release of inflammatory mediators, which may represent a new effective treatment strategy for ALI/ARDS [39].

#### 2.2.2. JAK2/STAT3 Signal Pathway

The Janus kinase 2 (JAK2)/signal transducer and activator of transcription 3 (STAT3) signaling pathway plays a key role in ALI/ARDS, and its activation is associated with IL-6 [40]. First, IL-6 binds to the receptor complex gp130/IL-6R on the surface of the host cell, thereby activating the JAK2/STAT3 signaling pathway. Then, the activated STAT3 dimer is transferred to the nucleus to initiate gene transcription [41], thereby regulating various activities (such as endothelial cell damage). Xu et al. [42] showed that in human pulmonary microvascular endothelial cells (HPMECs) and mouse macrophage Raw264.7 cell models induced by LPS, p-STAT3 inhibition reduces macrophage cytokine secretion and lung endothelial cell damage through the IL-6/JAK2/STAT3, NF-κB, and MAPK signaling pathways. Therefore, this pathway is involved in the occurrence and development of ALI/ARDS, and its regulation may provide effective treatment options for ALI/ARDS.

#### 2.2.3. The Role and Signaling Pathway of the NLRP3 Inflammasome in ALI/ARDS

NLRP3 belongs to the NLR family and is the most studied NLR. The classical pathway of NLRP3 inflammasome activation depends on the TLR4 signaling pathway and is mediated by the interaction of the first signal provided by PAMPs and the second signal provided by DAMPs. The first signal is mediated by microbial molecules or endogenous cytokines and then activates the TLR4/NF-κB signaling pathway, which subsequently upregulates the expression of NLRP3, pro-IL-18, and pro-IL-1β. The second signal is mediated by DAMPs; after NLRP3 recognizes PAMPs or DAMPs, it is oligomerized and recruits connexin ASC and caspase-1 precursors (pro-Caspase-1) for assembly into NLRP3 inflammasome, which leads to the maturation and activation of pro-Caspase-1 by self-cleavage [43]. Activated Caspase-1 also rapidly activates the precursors of IL-1β and IL-18 [44], leading to the maturation and release of IL-1β and IL-18.

The NLRP3 inflammasome plays an important role in ALI/ARDS. Studies have found that regulating the activation of NLRP3 can affect the process of ALI/ARDS. For example, morin, a flavonoid that exhibits significant antioxidant and anti-inflammatory activity, can inhibit the activation of NLRP3 inflammasomes and further prevent LPS-induced ALI [45]. Melatonin directly inhibits the activation of NLRP3 inflammasomes and improves ALI/ARDS by reducing the release of extracellular histones [46]. Additionally, pirfenidone can improve LPS-induced lung inflammation by blocking the activation of NLRP3 inflammasomes and subsequent secretion of IL-1β [47]. These results indicate that these agents could be used as potential therapeutics.

#### 2.2.4. PI3K/AKT Signaling Pathway

Recent studies have found that the PI3K/AKT signaling pathway is involved in the entire course of ALI/ARDS, including its etiology and pathogenesis. It is involved in early inflammation, pulmonary edema, and later tissue repair, airway remodeling, and emphysema [48]. On one hand, studies have shown that the phosphorylation levels of PI3K, AKT and mTOR were increased in LPS-induced ALI. Capsaicin (cap) is protective against LPS-induced ALI, which may be attributed to cap-mediated reduction of proinflammatory cytokines through inhibition of HMGB1/NF-κB and PI3K/AKT/mTOR pathways [49]. The activation of PI3K/AKT signaling pathway is consistent with the previous finding that cap induces autophagy and apoptosis in human nasopharyngeal carcinoma cells by downregulating the PI3K/AKT/mTOR pathway [50]. Similarly, studies have demonstrated that dexmedetomidine has a protective effect on LPS-induced ALI in vitro and in vivo, possibly by inhibiting inflammatory responses through HMGB1-mediated TLR4/NF-κB and PI3K/AKT/mTOR pathways [51]. The PI3K/AKT/mTOR and TLR4/NF-κB signaling pathways coordinate with each other in the inflammatory response [52], and NF-κB is a downstream target of AKT that activates NF-κB, leading to the translocation of p65 to the nucleus and the expression of proinflammatory genes [53]. On the other hand, Qu et al. found that the expression of PI3K/AKT signaling pathway was downregulated in LPS-induced ALI in in vitro and in vivo experiments. Further studies found that glycyrrhizic acid induced autophagy by inhibiting the activation of the PI3K/AKT/mTOR pathway, thereby improving LPS-induced ALI [54]. Deng et al. [55] also reported that the PI3K/AKT signaling pathway was downregulated in an LPS-induced ALI rat model; however, they found that PI3K/AKT signaling pathway activation can enhance the activity of alveolar sodium channels and Na^+^-K^+^-ATP, thereby removing the excess edema fluid and reducing the exudation of protein-rich fluid in the alveoli, which ultimately reduces lung tissue damage [56]. Another study also showed that insulin could reduce LPS-induced pulmonary edema in ALI rats, enhance alveolar fluid clearance and alleviate lung injury by activating PI3K/AKT signaling pathway, inhibiting Nedd4-2, and increasing ENaC expression [57]. However, these differences also suggest that the activation or inhibition of the PI3K/AKT pathway may be related to the time point selected by the injury model. The activation of the PI3K/AKT pathway may occur in the early stage, and the PI3K/AKT pathway may have a synergistic relationship with different pathways, especially the NF-κB signaling pathway.

#### 2.2.5. p38 MAPΚ Signaling Pathway

IL-1β, TNF-α, and other cytokines are chiefly produced in ALI/ARDS via the p38 MAPΚ signaling pathway [58]. Therefore, blocking the p38 MAPK signaling pathway can reduce the inflammatory response to some degree. For example, blocking the p38 MAPΚ signaling pathway with the p38 MAPK inhibitor SB203580 reduces excessive lung inflammation by inhibiting cell death caused by inflammatory macrophage apoptosis [58]. Epicatechin (EC), one of several anti-inflammatory interventions recently described, is also involved in LPS-induced ALI in a mouse model. EC may directly bind to the active site of p38 and inhibit its catalytic activity, thereby inhibiting inflammatory damage by inhibiting activation of the p38 MAPK-AP-1 signaling pathway [59]. Additionally, taurine can reduce ALI/ARDS caused by sepsis by inhibiting the p38/MAPK signaling pathway as well as inflammation and oxidative stress [60]. Thus, the p38/MAPK signaling pathway is being investigated as a potential therapeutic target for ALI/ARDS.

## 3. The Role of MicroRNAs in LPS-Induced ALI/ARDS

MicroRNAs chiefly bind to the 3′-UTR of mRNA to degrade or suppress the expression of target mRNA (Figure 2), which play a role in the physiological and pathological functions of a variety of disease processes. As previously stated, miRNAs play an important role in inflammation, and inflammation is an important characteristic of ALI/ARDS, miRNAs were proposed as diagnostic indicators and therapeutic targets for ALI/ARDS [24]. Indeed, studies have shown that miRNAs regulate ALI/ARDS via different mechanisms (Figure 3).

### 3.1. Biogenesis and Biological Function of MiRNAs

Most miRNAs are initially transcribed by RNA polymerase II [61] in the nucleus into primary miRNA (pri-miRNA) with cap structure (7MGpppG) and polyadenylic acid tail (AAAAA) (Figure 2). In the classical pathway, the conversion of pri-miRNA into mature miRNA requires two steps, which are catalyzed by Drosha and Dicer, two members of the RNase III family. In the first step, these pri-miRNAs are processed into 60 to 100 nt RNAs with a stem-loop structure by the RNase III endonuclease Drosha as well as its cofactor DiGeorge syndrome critical region 8 (DGCR8) in the nucleus [62,63], termed pre-miRNAs. A single pri-miRNA can produce one or more pre-miRNAs. After pre-miRNAs are transported into the cytoplasm via Exportin-5, the second phase of miRNA biosynthesis [64] is initiated in the cytoplasm. In the cytoplasm, pre-miRNAs are cut by the second RNase III endonuclease Dicer, producing a mature miRNA duplex (miRNA-miRNA*) [65,66]. Subsequently, the miRNA duplex is loaded into the Argonaute (AGO) protein, together with Dicer and TAR RNA binding protein (TRBP), a known chaperone of the miRNA processing enzyme Dicer that alters the rate of pre-miRNA cleavage in an RNA structure-specific manner [67], to form the RNA-induced silencing complex (RISC). The miRNA strand in the miRNA complex is retained in RISC, whereas the miRNA* strand is degraded. Then, the RISC binds to the 3′ or 5′ UTR, open reading frame (ORF), or promoter region of the target mRNA [68]. As a key regulator of human miRNA processing and targeting, TRBP is required for the recruitment of AGO2 to Dicer-bound small interfering RNA (siRNA) [66] Furthermore, TRBP can trigger the production of iso-miRNAs (isomiRs) that are one nucleotide longer than the standard sequence. It has been found that alterations in miRNA processing sites can alter guide strand selection, resulting in preferential silencing of different mRNA targets [67]. As the catalytic engine of RISC, AGO proteins, especially AGO2, mediate the process of miRNA-induced gene silencing, including mRNA degradation and translation inhibition [69]. For example, miR-223, a miRNA that is located on the X chromosome, is deregulated in various pathogenic conditions, such as type II diabetes, ALI, rheumatoid arthritis, and inflammatory bowel disease [70,71]. miR-223 specifically targets the 3′-UTR of NLRP3 mRNA, thereby inhibiting the translational expression of NLRP3 [72]. Therefore, NLRP3 expression can be controlled by regulating the expression of miR-223. In addition, miRNAs also regulate transcription factors through methylation, thus indirectly altering gene expression. For example, methylation of the miR-495 promoter region can downregulate its expression, contributing to the activation of the NLRP3 inflammasome and ultimately resulting in ALI/ARDS [73].

To date, greater than 2000 miRNA genes have been identified in the human genome [74]. Of note, each miRNA can target hundreds of mRNAs, and a single mRNA can be targeted by multiple miRNAs [75,76]. In addition, the expression of miRNAs has characteristics of tissue and cell specificity and time and space specificity, and miRNAs can represent useful clinical biomarkers [77]. For instance, miR-122, which is highly specifically expressed in the liver, is considered a promising biomarker of liver damage [78]. Additionally, miR-199b-3p, miR-1975, miR-15b, and miR-421 are considered to be biomarkers for liver cancer diagnosis [79], and miR-7, miR-9-5p, miR-9-3p, miR-129, and miR-132 [80] can be used as noninvasive methods to diagnose Parkinson’s disease. Zhu et al. [81] found that miR-652-3p can directly regulate KCNN3 to promote the proliferation, migration, and invasion of bladder cancer cells, and it is likely to be a potential target for bladder cancer treatment. In addition, some miRNAs, such as miR-150, miR-122, and miR-25, have been confirmed to be abnormally expressed in patients with sepsis and may be used as biomarkers for the early diagnosis and prognosis of sepsis [79]. These studies have laid a foundation for miRNA as a biomarker of ALI/ARDS.

### 3.2. Adverse Inflammation-Related MiRNAs

Some miRNAs can regulate ALI/ARDS by activating inflammatory responses. Inflammation-related miRNAs, such as miR-34a, miR-132, miR-155, miR-15a, miR-21, miR-27b, and miR-146a, were differentially expressed in lung tissues in an LPS-induced ALI model [82,83]. These miRNAs may regulate ALI/ARDS by activating inflammatory responses.

For example, miR-34a regulates alveolar epithelial cell function by targeting Forkhead box O3 (FOXO3) (Table 1) [84] or Angiotension 1 (Ang1) [85] to mediate the NF-κB signaling pathway [86]. Similarly, miR-326 and miR-300 also stimulate the NF-κB signaling pathway. The difference is that miR-326 targets B-cell lymphoma 2-related A1 (BCL2A1) [87] in the LPS-induced AEC model, and miR-300 targets the inhibitor κBα (IκBα) protein in A549 cells treated with LPS [88]. In a model of ALI induced by staphyloccocal enterotoxin B (SEB), Rao et al. [89] found that miR-132 was upregulated, resulting in the inhibition of FOXO3 and the activation of NF-κB signaling pathway [90], which subsequently leads to an overactivation of the immune response, exacerbating inflammatory damage.

MiR-155, the main regulator of lymphocyte development, differentiation, and function, is initiated in macrophages/monocytes via the TLR2, TLR3, TLR4, or TLR9 signaling pathway and negatively regulates inflammatory responses by targeting MyD88, TAB2, IKKε, and/or other TLR signaling pathway components [91,92]. miR-155 derived from serum exosomes can promote inflammation in ALI/ARDS by targeting suppressor of cytokine signaling 1(SOCS1) [93], a known target of miR-155 that can inhibit NF-κB activity by reducing the stability of p65. Additionally, miR-155 can promote macrophage proliferation by targeting SHIP1, a tumor suppressor that can promote macrophage proliferation [93]. In addition, miR-155 also showed the ability to inhibit the M2 special factor C/EBPb and IL-13R, thereby inhibiting the development of alternately activated macrophages. This finding indicates that miR-155 mutually exclusively regulates M1/M2 polarization [94]. Because M2 macrophages are mediated by the secretion of IL-10 and induce the expansion of CD4^+^ CD25^+^ regulatory T cells (Tregs), miR-155 antisense oligonucleotide (ASO) treatment can promote the recovery of ALI [95].

Goodwin et al. [96] found that miR-887-3p alters endothelial cell function in several important ways, including increased cytokine and neutrophil chemokine release and enhanced trans-endothelial migration, exacerbating the development of ARDS immune inflammation. In addition, studies have pointed out that inhibiting the expression of miR-34a-5p or miR-1246 can inhibit oxidative stress, reduce endothelial cell apoptosis, and reduce the expression level of inflammatory cytokines [86]. Furthermore, inhibiting the expression of miR-92a, miR-34b-5p, and miR-199a can reduce excessive inflammation and apoptosis while increasing the survival rate of ALI/ARDS mice [86]. In LPS-induced ALI mice, miR-203 can suppress the PI3K/AKT signaling pathway by inhibiting PIK3CA, a gene encoding a PI3K catalyst, to increase the downstream apoptotic proteins, and ultimately inhibit epithelial cell proliferation and promote apoptosis [97,98]. Furthermore, repressed miR-203 effectively attenuated LPS-induced interstitial pneumonia.

### 3.3. Protective Inflammation-Associated MiRNAs

#### 3.3.1. Targeting ALI/ARDS through NF-κB Signaling Pathway

For the past few years, a series of studies have demonstrated that miRNAs can influence the pathological process of ALI/ARDS via TLR4/NF-κB signaling pathway. TLR4 expression is increased under LPS stimulation and could be inhibited by some miRNAs, which ultimately alter the inflammatory response. For example, miR-27a [99,100,101] and miR-16 [102] can inhibit the expression of TLR4 by targeting its 3′-UTR, reducing proinflammatory cytokine release and inflammatory responses by inhibiting downstream TLR4/MyD88/NF-κB signaling pathways [100]. Therefore, upregulating the expression of miR-27a and miR-16 is beneficial to suppress the inflammatory response. Similarly, in the LPS-induced ALI model, Yang et al. [103] found that miR-182 inhibits the activation of NF-κB by targeting the expression of TLR4, negatively regulates the TLR4/NF-κB pathway, and improves LPS-induced ALI. Furthermore, overexpression of miR-145-5p, miR-16, miR-140, and miR-140-5p can inhibit TLR4 expression and block NF-κB pathway activation, thereby alleviating ALI/ARDS [86]. Additionally, miR-146a participates in TLR4/NF-κB signal transduction by inhibiting IRAK1 and TRAF6 [104], thus reducing proinflammatory factors, such as TNF-α, IL-6, and IL-1β, in alveolar macrophages. He et al. [105] found that miR-146b reduces lung inflammation and increases lung permeability by targeting IRAK1 to inhibit NF-κB signaling. Therefore, miR-146b overexpression could alleviate LPS-induced ALI. In the LPS-induced mouse ARDS model, miR-124-3p targets p65, promotes macrophage apoptosis, and plays a protective role in ARDS [106]. Sun et al. [107] suggested that miR-181b inhibits the NF-κB signaling pathway mainly by directly targeting importin-α3, a key protein in nuclear translocation of NF-κB. In contrast, miR-181b was detected to be elevated after LPS treatment of Beas-2b cells for 48 h [108]. Overexpressed miR-181b increases p65 and activates the NF-kB signaling pathway [109]. The changes and roles of miR-181b in lung injury need further experimental verification.

#### 3.3.2. Targeting ALI/ARDS through JAK2/STAT3 Signaling Pathway

MiRNAs can affect the pathological process of ALI/ARDS through the JAK2/STAT3 signaling pathway, and JAK2/STAT3 inhibitor intervention can reverse the proinflammatory response induced by ALI/ARDS. Studies have confirmed that miR-21 expression is inhibited and JAK2/STAT3 and NF-κB signal transduction is activated in LPS-induced ALI/ARDS [110]. Upregulation of miR-21 can inhibit the JAK2/STAT3 signaling pathway, thereby reducing the infiltration of inflammatory cells in the lung tissue of ALI/ARDS mice induced by LPS [110]. Similarly, Kong et al. [111] noted that miR-216a expression was sharply reduced in ALI/ARDS patients and overexpression of miR-216a can inhibit the JAK2/STAT3 signaling pathway, inhibiting cell apoptosis, autophagy, and the release of inflammatory factors [111], thereby reducing LPS-induced ALI/ARDS. In addition, miR-30b-5p can bind to the 3′-UTR of suppressor of cytokine signaling 3 (SOCS3) [112], negatively regulating the JAK2/STAT3 pathway that mediates inflammation of lung macrophages and inhibiting the expression of ALI/ARDS inflammatory factors [113]. In addition, in a lung injury model in vivo, miR-127 was found to be downregulated, and its overexpression weakened lung permeability, inflammatory cell infiltration, cytokines, complement components, and STAT3 activation and signal transduction [114]. Further research found that miR-127 suppresses lung inflammation by targeting macrophage CD64 [114]. In contrast to this study, studies have shown that miR-127 targets B-cell lymphoma 6 (BCL6), which subsequently reduces the expression of dual-specific phosphatase 1 (DUSP1), and the JNK kinase and P38 MAPK signaling pathways are subsequently activated. This ultimately results in the formation of proinflammatory macrophages and a significant increase in proinflammatory cytokines [94]. However, these differences also demonstrate that miRNAs and diseases are related to time, cells, and damage models.

#### 3.3.3. Targeting ALI/ARDS by Regulating NLRP3 Signaling Pathway

NLRP3 is linked to the occurrence and progression of pulmonary inflammatory diseases [115]. Overexpression of miR-495 inhibits the development of ALI/ARDS by negatively regulating NLRP3. This finding indicates that miR-495 upregulation could serve as a promising therapeutic target for ALI/ARDS treatment [73]. Additionally, You et al. [116] confirmed that miR-802 can improve lung injury induced by LPS by targeting pellino E3 ubiquitin-protein ligase family member 2 (Peli2), which mediates the activation of NLRP3 by promoting its ubiquitination [117]. Evidence indicates that the activation of NLRP3 inflammasome can be inhibited by miR-223 [73,118], which consequently reduces inflammation and improves ALI/ARDS.

#### 3.3.4. Targeting ALI/ARDS through PI3Κ/AKT Signaling Pathway

Previous research has shown that the PI3K/AKT signaling pathway is significant in the regulation of ALI/ARDS by miRNAs. For example, miR-92a expression is increased in the HPMEC model induced by LPS [119]. Inhibition of miR-92a expression can significantly increase the migration of HPMECs, enhance the formation of blood vessels, and improve endothelial cell barrier dysfunction [119]. A further study found that miR-92 can target integrin α5 (ITGA5), thereby inhibiting the PI3K/AKT signaling pathway. Therefore, miR-92a inhibitors can act as protective agents for alveolar vascular endothelial cells [100]. Zhou et al. [120] found in the LPS-induced mouse ARDS model that miR-21a-3p in stromal cell Telocytes (TCs) could regulate the PI3K (p110α)/Akt/mTOR pathway to promote lung tissue repair and angiogenesis, which is beneficial to recovery of ARDS.

**Table 1 ijms-23-05545-t001:** Targets and function of microRNA in ALI/ARDS.

Type	MicroRNA	Damage Factors	Target	Expression	Function	Signaling Pathway	Reference
Adverse	miR-34a	LPS	FOXO3	upregulation	Proinflammatory	NF-κB signaling pathway	[84]
	miR-34a	hyperoxia	Ang1	upregulation	Proinflammatory	NF-κB signaling pathway	[85]
	miR-326	LPS	BCL2A1	upregulation	Proinflammatory	NF-κB signaling pathway	[87]
	miR-300	LPS	IκBα	upregulation	Proinflammatory	NF-κB signaling pathway	[88]
	miR-132	SEB	FOXO3	upregulation	Proinflammatory	NF-κB signaling pathway	[90]
	miR-155	LPS	SOCS1	upregulation	Proinflammatory	NF-κB signaling pathway	[93]
	miR-155	LPS	C/EBPb and IL-13R	upregulation	Proinflammatory	regulates M1/M2 polarization	[94]
	miR-887-3p	LPS		upregulation	Proinflammatory		[96]
	miR-34a-5p	LPS and hyperoxia	SIRT1	upregulation	Proinflammatory		[121]
	miR-1246	LPS	ACE2	upregulation	Proinflammatory		[122]
	miR-92a	LPS	ITGA5	upregulation	Proinflammatory	PI3Κ/AKT signaling pathway	[119]
	miR-34b-5p	LPS	PGRN	upregulation	Proinflammatory		[123]
	miR-199a	LPS	SIRT1	upregulation	Proinflammatory		[124]
	miR-181b	LPS	P65	upregulation	Proinflammatory	NF-κB signaling pathway	[108,109]
	miR-127	LPS	BCL6	upregulation	Proinflammatory	P38 MAPK signaling pathway	[94]
Protective	miR-27a	LPS	TLR4	downregulation	anti-inflammatory	NF-κB signaling pathway	[99,100]
	miR-16	LPS	TLR4	downregulation	anti-inflammatory	NF-κB signaling pathway	[102]
	miR-182	LPS		downregulation	anti-inflammatory	NF-κB signaling pathway	[103]
	miR-145-5p	LPS	TLR4	downregulation	anti-inflammatory	NF-κB signaling pathway	[125]
	miR-140	LPS	TLR4	downregulation	anti-inflammatory	NF-κB signaling pathway	[126]
	miR-140-5p	LPS	TLR4	downregulation	anti-inflammatory	NF-κB signaling pathway	[127]
	miR-146a	LPS	IRAK1 TRAF6	downregulation	anti-inflammatory	NF-κB signaling pathway	[104]
	miR-146b	LPS	IRAK1	downregulation	anti-inflammatory	NF-κB signaling pathway	[105]
	miR-181b	LPS	importin-α3	downregulation	anti-inflammatory	NF-κB signaling pathway	[107]
	miR-124-3p	LPS	p65	downregulation	anti-inflammatory	NF-κB signaling pathway	[106]
	miR-21	LPS	JAK2	downregulation	anti-inflammatory	JAK2/STAT3 and NF-κB signaling pathways	[110]
	miR-216a	LPS	JAK2	downregulation	anti-inflammatory	JAK2/STAT3 signal pathway	[111]
	miR-30b-5p	LPS	SOCS3	downregulation	anti-inflammatory	JAK2/STAT3 signal pathway	[112]
	miR-127	LPS	CD64	downregulation	anti-inflammatory	STAT3 signaling pathway	[114]
	miR-495	LPS	NLRP3	downregulation	anti-inflammatory	NLRP3 signaling pathway	[73]
	miR-802	LPS	Peli2	downregulation	anti-inflammatory	NLRP3 signaling pathway	[117]
	miR-223	LPS	NLRP3	downregulation	anti-inflammatory	NLRP3 signaling pathway	[73,118]
	miR-21a-3p	LPS		downregulation	anti-inflammatory	PI3K (p110α)/Akt/mTOR pathway	[120]

**FoxO3**: Forkhead box O3; **Ang1**: Angiotension 1; **BCL2A1**: B-cell lymphoma 2-related A1; **IκBα**: inhibitor κBα; **SOCS1**: suppressor of cytokine signaling 1; **SIRT1**: Sirtuin 1; **ACE2**: Angiotensin-converting enzyme 2; **ITGA5**: integrin α5; **SOCS3**: suppressor of cytokine signaling 3; **IRAK1**: IL-1 receptor associated kinase; **TRAF6**: Tumor necrosis factor receptor-associated factor; **JAK2**: Janus kinase 2; **BCL6**: B-cell lymphoma 6; **Peli2**: pellino E3 ubiquitin-protein ligase family member 2.

### 3.4. Acute Lung Injury/Acute Respiratory Distress Syndrome and Exosomal MicroRNAs

Exosomes, secreted microvesicles that transport miRNAs, mRNAs, and proteins via humoral transport, facilitate intercellular communication and initiate immune responses. The content of exosomes varies with cellular origin and physiological conditions, and can provide insights into how cells and systems respond to physiological perturbations.

McDonald et al. [128] demonstrated that LPS-stimulated macrophage-secreted exosomes carried higher levels of three mouse homologous human miRNAs (miR-21-3p, miR-146a and miR-146b) with known roles in preventing overactivation of the innate immune response (Figure 4). This results in inhibition of transcription and translation of proinflammatory cytokines. Multiple studies have shown that exosomes transfected with miR-145 and miR-223/142 can target mouse alveolar macrophages, reduce the secretion of inflammatory cytokines IL-2 and TNF-α, and inhibit the activation of NLRP3 inflammasome. Thus, inflammation can be reduced and sepsis-ALI can be improved [16,129,130]. Related studies have also shown that MSC-derived exosomes carrying miR-30b-3p can inhibit the apoptosis of AEC in LPS mice by reducing the level of human serum amyloid A3, and play a protective role in the lung of ALI mice [130]. Conversely, in vitro and in vivo experiments have shown that miR-155 derived from macrophage exosomes mentioned above has a proinflammatory effect by activating the NF-κB signaling pathway to induce inflammatory responses [93].

In addition, Quan et al. [131] found that exosomes derived from alveolar progenitor Type II cell (AT II C) multipotent stem cells carrying miR-371b-5p can improve the survival probability and even promote the proliferation of AT II C and promote the regeneration of damaged alveolar epithelium. Furthermore, endothelial progenitor cell-derived exosomes are enriched in miR-126, which can target and downregulate Sprouty-related EVH1 domain-containing protein 1 (SPRED1) and activate rapidly accelerated fibrosarcoma/extracellular signal regulated kinase (RAF/ERK) signaling pathway, improve endothelial cell function, enhance endothelial cell proliferation, migration, and tubule formation, and improve LPS-induced ALI and restoring lung integrity in vivo [132,133]. Furthermore, in ALI, the release of various proinflammatory factors by M0-Exosome vesicles (EVs) in bronchoalveolar lavage fluid is mainly damaged in the early stages. This activates neutrophils to produce IL-10, which may be responsible for polarizing M0 to M2c, leading to fibrosis after ALI [134].

In conclusion, exosomes possess systemic signaling capabilities, induce pleiotropic effects, and have great therapeutic potential.

## 4. The Potential Role of MiRNAs in Clinical Treatment of ALI/ARDS

Since the discovery of miRNAs in 1993, the field has progressed rapidly, and great achievements have been made in various aspects. Insights into the role of miRNAs in development and disease have also made miRNAs attractive tools and targets for novel therapeutic approaches.

In cancer diseases, for example, miRNAs function as tumor suppressor-miRs or oncogenes (OncomiRs), and miRNA mimics and miRNA-targeting molecules (anti-miRs) have shown promise in preclinical development [135]. In view of the many-to-many relationship between miRNAs and target genes, the targets selected in current clinical trials are generally proven to target multiple oncogenes or tumor suppressor genes [136], and miRNAs that function as Onco-miRs or Suppressor-miRs in a variety of malignant tumors [137]. Several miRNA-targeted therapies are already in clinical development. For example, Zhang et al. [138] found that the combination of six miRNAs (miR-21-5p, miR-20a-5p, miR-103a-3p, miR-106b-5p, miR-143-5p and miR-215) in tissues could be used to predict the effect of new treatment and long-term prognosis of stage II colon cancer patients. Wiemer et al. [139] conducted a clinical trial (NCT00413192) (Table 2) and found that 26 miRNAs could be used to predict the efficacy of the microtubule dynamics inhibitor eribulin in the treatment of metastatic soft tissue sarcoma. Miravisen, which is also known as SPC3649, is an antagonist of miR-122 [78]. It is the world’s first microRNA drug tested in humans as a new treatment for hepatitis C. Upon hepatitis C virus infection, the level of miR-122 is elevated. Unlike typical miRNAs, miR-122 activates mRNA, thereby enhancing the expression of hepatitis C-related genes and aggravating infection. Therefore, inhibiting the activity of miR-122 can play a therapeutic role. At present, Miravisen has passed Phase IV clinical trials [140], demonstrating that it is safe in the human body and indeed reduces the expression level of the hepatitis C virus. Other miRNA drugs for tumors such as antagonists of miR-34, let-7, and miR-16 are still in progress. Regarding the development of miRNA mimic-based cancer therapy, the most advanced compound is MRX34, a miR-34 mimic encapsulated in a lipid carrier called NOV40 [141], which has reached Phase I clinical trials for the treatment of cancer [141,142]. In mice treated with MRX34 nanoparticles, accumulation of miR-34 in tumors was observed, as well as significant tumor regression [141,143,144,145].

There are relatively few clinical trials targeting Onco-miRs, including one on the cancer-promoting molecule miR-155 (NCT02580552), in which researchers designed and synthesized a locked nucleic acid (LNA) called Cobomarsen or MRG-106 for the treatment of cutaneous T-lymphoma, Alibell’s disease, chronic lymphocytic leukemia and adult T-cell leukemia and lymphoma. The test has now been completed [146].

The application of miRNAs has also been studied in other diseases, such as cardiovascular disease. In an African green monkey model highly related to humans, antisense oligonucleotide inhibition of miR-33a/b was found to be a promising therapeutic strategy. It increases plasma high-density lipoprotein (HDL) and decreases very low-density lipoprotein (VLDL)-related triglyceride levels, used to treat dyslipidemia that increases the risk of cardiovascular disease, such as atherosclerosis [147]. Similarly, anti-miR-144 may also be a new approach to target HDL and reverse cholesterol transport [148]. Currently, the application of miRNAs in cardiac diseases is mostly focused on the biomarkers of disease occurrence and progression [149]. With further research, the first miR-132 inhibitor (CDR132L) for heart failure has entered clinical trials [149]. The success of this clinical trial marks the official entry of miRNA in the treatment of heart disease, setting a precedent for other miRNAs entering the clinic in the future for the treatment of heart disease [149]. In addition, studies have shown that miR-195-5p promotes cardiac hypertrophy by targeting MFN2 and FBXW7, which may provide a promising therapeutic strategy for intervening cardiac hypertrophy [150].

**Table 2 ijms-23-05545-t002:** MicroRNAs in clinical trials.

Name (Company)	Therapeutic Agent	Target Diseases	Trial Details	Clinical Trials.Gov Identifier	Reference
Mirvirasen (Santaris Pharma A/S and Hoffmann-La Roche)	Anti-miR-122	Hepatitis C (chronic infections included)	Phase III, completed	NCT01728324	[78,135]
			Phase III, completed	NCT01366638	
			Phase IV, completed	NCT01222611	
			Phase II, completed	NCT00996476	
			Phase III, completed	NCT01725529	
			Phase III, completed	NCT01290731	
			Phase III, completed	NCT01292239	
			Phase III, completed	NCT01288209	
		metastatic soft tissue sarcoma	Phase II, completed	NCT00413192	[139]
Cobomarsen or MRG-106 (miRagen Therapeutics)	Anti-miR-155	cutaneous T-lymphoma, Alibell’s disease, chronic lymphocytic leukemia and adult T-cell leukemia and lymphoma	Phase I, completed	NCT02580552	[146]
CDR132L	miR-132 inhibitor	heart disease	Phase I, completed	NCT04045405	[149]
MRX34 (Mirna Therapeutics)	miR-34 mimic	multiple solid tumours	Multicentre phase I, terminated	NCT01829971	[141,142,143,144,145]
MesomiR-1 (EnGeneIC)	miR-16 mimic	mesothelioma, non-small cell lung cancer	Phase I, completed	NCT02369198	[135]
		Acute Lung Injury/Acute Respiratory Distress Syndrome (ARDS)	Recruiting	NCT03766204	

At present, clinical trials using miRNAs as therapeutic targets focus on tumors, such as lung cancer, liver cancer and other diseases, and there are almost no clinical trials of miRNA for the diagnosis or treatment of ALI/ARDS. In the Clinical Trial website, a clinical experiment named “The Role of Circulating CircRNAs and MicroRNAs in Acute Lung Injury” (NCT03766204) is being carried out by Changhai hospital. The trial is currently recruiting and is expected to screen several non-coding RNAs as new biomarkers for ALI/ARDS. Therefore, clinical trials of miRNAs as therapeutic targets in ALI/ARDS are urgently needed, and the clinical application of miRNAs in ALI/ARDS deserves attention. As miRNAs play important roles in the process of ALI/ARDS, they have great prospects in the clinical treatment of ALI/ARDS. With the proliferation of human genome data and the development of new delivery vectors, we are optimistic that the use of miRNAs as therapeutic targets for ALI/ARDS will become a clinical reality.

## 5. Summary and Future Prospects

Due to the complex etiology of ALI/ARDS, after half a century, although significant progress has been made in the pathogenesis of ALI/ARDS, the exact pathogenesis of ALI/ARDS has not yet been fully elucidated. At present, uncontrolled inflammation is generally believed to be the main pathophysiological change in ALI/ARDS [58,101,151]. Therefore, inhibiting the uncontrolled inflammatory response may be the key strategy for ALI/ARDS treatment. Evidence indicates that miRNAs play an essential role in regulating the inflammatory response process in ALI/ARDS. Therefore, exploring the miRNAs related to inflammatory signaling pathways in ALI/ARDS has very important clinical significance for the early diagnosis and treatment of diseases. This article describes the role of miRNAs in the pathogenesis of ALI/ARDS from miRNA-related inflammatory signaling pathways to provide new treatment ideas for ALI/ARDS.

Although research on the regulation of ALI/ARDS by miRNAs is still constantly being updated, the existing research conclusions are sufficient to demonstrate its importance in the occurrence and progression of ALI/ARDS. These regulatory effects exist at the level of cells, receptors, signaling pathways, and gene transcription [152]. Such a complex and delicate regulatory mechanism of miRNA makes the entire signaling pathway a promising therapeutic drug (usually in the form of miRNA analogs) or drug therapy targets (usually in the form of anti-miRNA) in several pathological processes.

Increasing evidence shows that changes in specific miRNA levels are correlated with various diseases, and miRNAs are considered as potential biomarkers for the diagnosis of various diseases, such as cancer and cardiovascular diseases. As mentioned above, miRNAs play an essential role in regulating the inflammatory response, the expression of miRNAs (such as miR-155, miR-27a, miR-21, miR-146a, and miR-223) is altered in ALI/ARDS, and some miRNAs have also been shown to be new biomarkers for the prognosis of ALI/ARDS. Consequently, miRNAs are also considered as potential therapeutic targets for diseases, and clinical trials of some miRNA drugs are underway.

In summary, this article describes the function of miRNAs in the pathogenesis of ALI/ARDS, and increasing the expression of protective miRNAs and inhibiting unfavorable miRNAs have been demonstrated to be effective in alleviating ALI/ARDS in many studies. Some miRNAs have also been shown to be biomarkers of ALI/ARDS; therefore, regulating the expression of miRNAs in ALI/ARDS may represent a future treatment direction for lung injury.

## Figures and Tables

**Figure 1 ijms-23-05545-f001:**
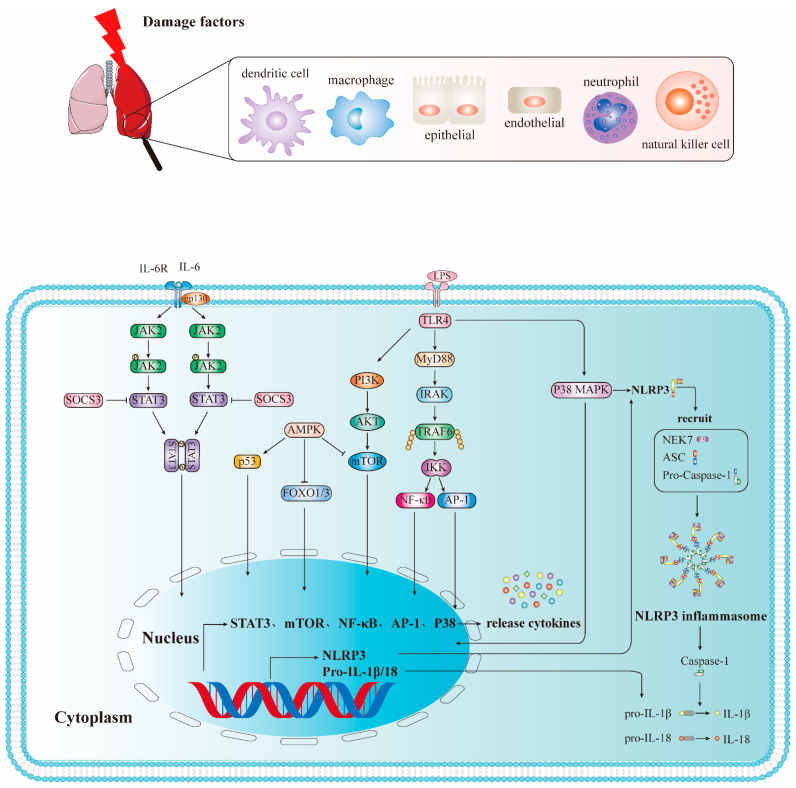
Inflammation-related pathways of ALI/ARDS. **DC**: dendritic cell; **NK**: natural killer cell; **JAK2**: Janus kinase 2; **STAT3**: signal transducer and activator of transcription 3; **SCOS3**: Suppressor of Cytokine Signaling 3; **PI3K**: phosphatidylinositol 3-kinase; **AKT**: protein kinase B (PKB, also known as AKT); **mTOR**: Mammalian Target of Rapamycin; **TLR4**: Toll-like receptor 4; **MyD88**: Myeloid differentiation factor 88; **IRAK**: interleukin-1 receptor-associated kinase; **TRAF6**: tumor necrosis factor receptor-associated factor 6; **IKK**: Ikappa B kinase; **NF-κB**: nuclear factor kappaB; **AP-1**: activator protein-1; **NLRP3**: NOD-like receptor thermal protein domain associated protein 3; **ASC**: apoptosis-associated speck-like protein containing a CARD; **Pro-Caspase-1**: pro-cysteinyl aspartate specific proteinase-1; **Caspase-1**: cysteinyl aspartate specific proteinase-1.

**Figure 2 ijms-23-05545-f002:**
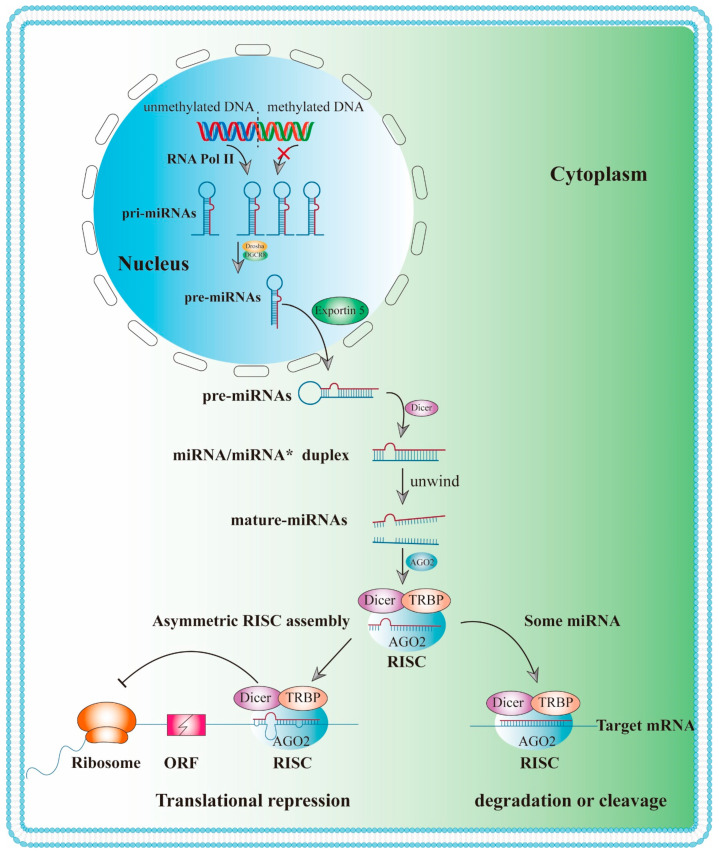
Biogenesis and biological function of miRNAs. **RNA Pol II**: RNA polymerase II; **DGCR8**: DiGeorge syndrome critical region 8; **pri-miRNAs**: primary miRNA; **pre-miRNAs**: microRNAs precursor; **RISC**: RNA-induced silencing complex; **TRBP**: TAR RNA binding proteins; **AGO2**: Argonaute 2; **ORF**: open reading frame.

**Figure 3 ijms-23-05545-f003:**
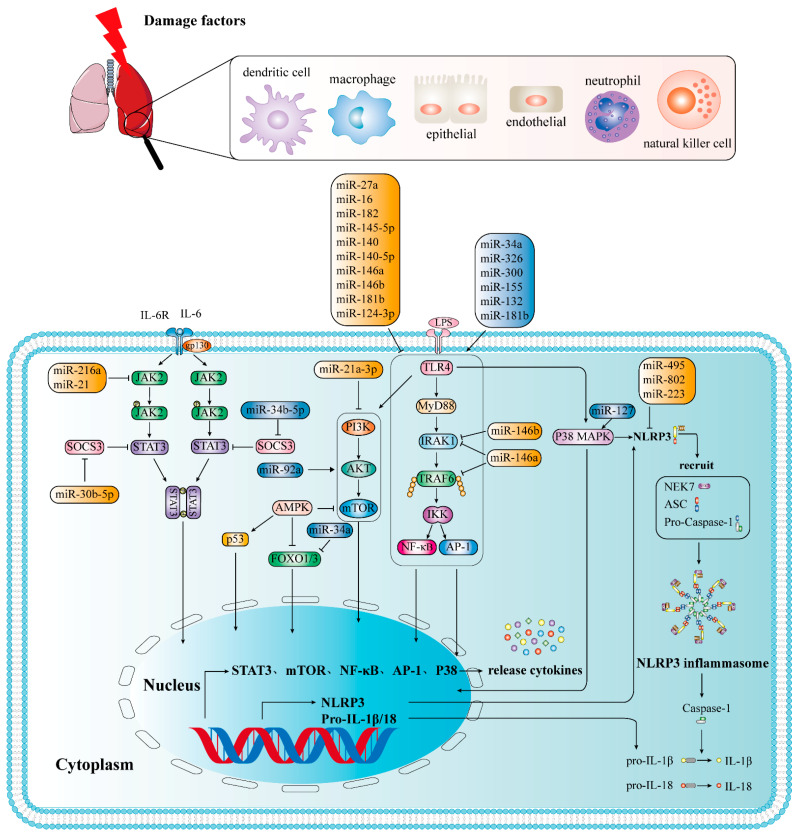
The role of microRNAs in LPS-induced ALI/ARDS.

**Figure 4 ijms-23-05545-f004:**
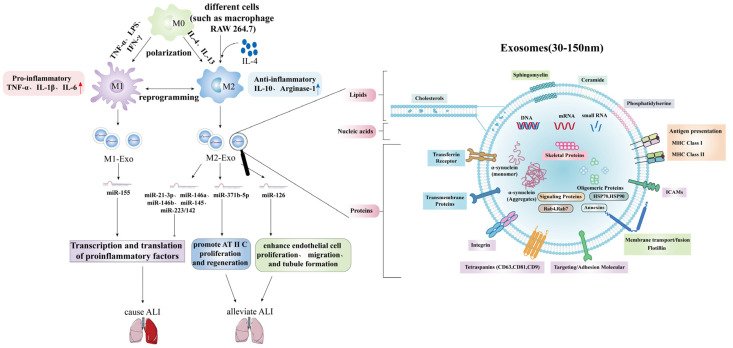
The role of exosomal microRNAs in LPS-induced ALI/ARDS.

## Data Availability

Not applicable.

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
