# Peer review of "MicroRNAs: Important Regulatory Molecules in Acute Lung Injury/Acute Respiratory Distress Syndrome"

_ijms, 2022, doi:10.3390/ijms23105545_

Round 1
Reviewer 1 Report
In the review manuscript titled, “MicroRNAs: Important regulatory molecules in acute lung injury/acute respiratory distress syndrome”, the authors provided an update on the functional role of microRNAs in the pathogenesis of ALI/ARDS and highlighted the potential role of miRNAs as biomarkers and as future therapeutic options for ALI/ARDS. The authors are recommended to make the following changes.
- In the introduction section, lines 28-30, authors should include other common causes of ALI/ARDS such as trauma, smoke inhalation.
- Published literature have reported that PI3K/AKT signaling pathway is also downregulated in LPS-induced ALI/ARDS. Authors should include these findings and provide a plausible explanation for the differential regulation of this pathway with LPS treatment.
- Authors should look up for recent clinical trials of microRNAs in ALI/ARDS and include in the review.
- Are there any drugs that inhibit or upregulate microRNAs? Authors should include them in the review, even these drugs are currently used for the treatment of other diseases.
- Authors are recommended to check the flow of the manuscript.
Author Response
Response to Reviewer 1 Comments
Point 1: In the introduction section, lines 28-30, authors should include other common causes of ALI/ARDS such as trauma, smoke inhalation.
Response 1: Thank you for your good suggestion. In the introduction section, common causes of ALI/ARDS include trauma, smoke inhalation have added as your mentioned. In addition, we have also added two relatively rare causes of ALI/ARDS, blast and asphyxiating agents. Please see the detailed modification in the introduction section.
Point 2: Published literature have reported that PI3K/AKT signaling pathway is also downregulated in LPS-induced ALI/ARDS. Authors should include these findings and provide a plausible explanation for the differential regulation of this pathway with LPS treatment.
Response 2: Thank you for your valuable and constructive suggestions. Based on your comments, we have reviewed and categorized the literature on PI3K/AKT signaling pathway regulation in ALI/ARDS. We included literature and drug intervention results on the up- and down-regulation of PI3K/AKT signaling pathway in LPS-induced ALI/ARDS, respectively. And we speculate on the reasons for the differential regulation of this pathway.
Point 3: Authors should look up for recent clinical trials of microRNAs in ALI/ARDS and include in the review.
Response 3: Thank you for your valuable and constructive suggestions. Based on your suggestion, we searched the website of the China Clinical Trial Center with the keyword "miRNA" and the clinical trial website with the keywords of "miRNAs" and "acute lung injury/acute respiratory distress syndrome (ARDS)", and found that Changhai hospital has conducted a clinical trial titled " The Role of Circulating CircRNAs and MicroRNAs in Acute Lung Injury " (NCT03766204). The trial is currently recruiting and is expected to screen several non-coding RNAs as new organisms for ALI/ARDS Markers. We have added the details of this part to Part 4.
Point 4: Are there any drugs that inhibit or upregulate microRNAs? Authors should include them in the review, even these drugs are currently used for the treatment of other diseases.
Response 4: As of the writing of this review, there are no miRNA-based drugs for the treatment of ALI/ARDS. However, clinical trials have begun evaluating miRNA-based therapies in other diseases. There are several strategies to modulate the level of miRNA expression, which are to increase miRNA expression by delivering miRNA mimics or expression vectors (usually plasmids or viruses). In contrast, anti-miRNA oligonucleotides (AMOs), miRNA sponges or specific miRNA small molecule inhibitors (SMIRs) inhibited miRNA expression[1].
We have made a summary of the miRNAs currently in clinical trials in Section 4 of the article. However, these drugs are mainly used in the treatment of hepatitis C, cancer, cardiovascular and other diseases, and have not yet been used in the treatment of ALI/ARDS. With the increase in human genomic data and the development of new delivery vehicles, we are optimistic that miRNAs as therapeutics or therapeutic targets for human diseases, including ALI/ARDS, will become a clinical reality.
Point 5: Authors are recommended to check the flow of the manuscript.
Response 5: Thank you for your valuable and constructive suggestions. We have greatly benefited from your comments and have made extensive revisions to the manuscript based on your suggestions. In order to make the content of the article more coherent, we have integrated the second, fourth, and fifth parts of the original manuscript into the second part, and exchanged the positions with the third part. Then, the third part of the original manuscript, "Injury Mechanisms of ALI/ARDS" was revised as " Mechanisms leading to tissue damage in ALI/ARDS ", and greatly deleted, so that readers can pay more attention to the main content of our article, that is, the important role of microRNAs in ALI/ARDS. In addition, we have added the potential role of miRNAs in the clinical treatment of ALI/ARDS, which is now in the fourth part of the article.
Reference
- Lee, L.K., et al., The Role of MicroRNAs in Acute Respiratory Distress Syndrome and Sepsis, From Targets to Therapies: A Narrative Review. Anesth Analg, 2020. 131(5): p. 1471-1484.

Reviewer 2 Report
The review article is focused on the role of microRNAs in ALI/ARDS. The article will be a good compilation of the literature on microRNAs mediating ALI for readers in this field. I have the following suggestions for improving the manuscript.
The review title is very specific for microRNAs in ALI/ARDS. However, there is a disconnect between the 'biogenesis of microRNAs' and the real discussions on microRNAs in ALI interrupted by the general mechanisms of ALI/ARDS. In fact, the discussions on miRs in lung injury do not start until Section 4. It is highly recommended to make the general mechanisms of lung injury very succinct and brief and move it before the biogenesis of microRNAs to maintain a good flow of information. There are a lot of review articles available that discuss mechanisms of ALI but not much on microRNAs. Hence, authors should focus on the sections more related to the topic without diluting it with other mechanisms. The mechanisms should be limited to the links that need to be created to explain their regulation by microRNAs in lung injury.
Subheading 3 "Damage mechanism of ALI/ARDS" may be revised as "Mechanisms leading to tissue damage in ALI/ARDS".
Section 6 can be revised and "Summary and future prospects"
Except for these, the review looks great!
Author Response
Response to Reviewer 2 Comments
Point 1: There is a disconnect between the 'biogenesis of microRNAs' and the real discussions on microRNAs in ALI interrupted by the general mechanisms of ALI/ARDS. In fact, the discussions on miRs in lung injury do not start until Section 4. It is highly recommended to make the general mechanisms of lung injury very succinct and brief and move it before the biogenesis of microRNAs to maintain a good flow of information. There are a lot of review articles available that discuss mechanisms of ALI but not much on microRNAs. Hence, authors should focus on the sections more related to the topic without diluting it with other mechanisms. The mechanisms should be limited to the links that need to be created to explain their regulation by microRNAs in lung injury.
Response 1: Thank you for your valuable and constructive suggestions. We have greatly benefited from your comments and have made extensive revisions to the manuscript based on your suggestions. In order to make the content of the article more coherent, we have integrated the second, fourth, and fifth parts of the original manuscript into the second part, and exchanged the positions with the third part. Then, the third part of the original manuscript, "Injury Mechanisms of ALI/ARDS", has been greatly deleted, so that readers can pay more attention to the main content of our article, that is, the important role of microRNAs in ALI/ARDS. In addition, we have added the potential role of miRNAs in the clinical treatment of ALI/ARDS, which is now in the fourth part of the article.
Point 2: Subheading 3 "Damage mechanism of ALI/ARDS" may be revised as "Mechanisms leading to tissue damage in ALI/ARDS".
Response 2: Thanks for your comment. We've edited the subheading as you suggested.
Point 3: Section 6 can be revised and "Summary and future prospects".
Response 3: Thanks for your comment. We have revised the heading of Section 6 (now is Section 5) as you suggested.

Round 2
Reviewer 1 Report
In the revised manuscript titled, “MicroRNAs: Important regulatory molecules in acute lung injury/acute respiratory distress syndrome” the authors have incorporated all the changes recommended by the reviewers. The authors are recommended to the make the following additional changes.
- The authors are recommended to upload the revised version after turning off the track changes. It is inconvenient to read the revised manuscript with the track changes on.
- In lines 586-588, “In LPS-induced ALI mice, miR-203 can activate the PI3K/AKT signaling pathway by inhibiting PIK3CA, a gene encoding a PI3K catalyst, to increase the downstream apoptotic proteins, and ultimately inhibit epithelial cell proliferation and promote apoptosis”, the authors should check whether miR-203 inhibits or activates PI3K/AKT signaling. If PIK3CA gene is inhibited the pathway should be downregulated.
Author Response
Response to Reviewer 1 Comments
Point 1: The authors are recommended to upload the revised version after turning off the track changes. It is inconvenient to read the revised manuscript with the track changes on.
Response 1: Thanks for your comment. The journal requires that “Any revisions made to the manuscript should be marked up using the“Track Changes” function, such that changes can be easily viewed by the editors and reviewers.”, so in the last revision, we revised it using the“Track Changes”. Since the structure of the manuscript of our previous draft has been substantially revised, it is true it is inconvenient to read the revised manuscript with the track changes on. We are very sorry for this. We will upload two versions this time, “manuscript-track changes” for the easy viewing of changes, and “manuscript-final version” for the convenience of viewing the full text. Please check our uploaded zip file “manuscript revision”.
Point 2: In lines 586-588, “In LPS-induced ALI mice, miR-203 can activate the PI3K/AKT signaling pathway by inhibiting PIK3CA, a gene encoding a PI3K catalyst, to increase the downstream apoptotic proteins, and ultimately inhibit epithelial cell proliferation and promote apoptosis”, the authors should check whether miR-203 inhibits or activates PI3K/AKT signaling. If PIK3CA gene is inhibited the pathway should be downregulated.
Response 2: Thank you for your valuable and constructive suggestions. Based on your suggestion, we checked, confirmed and corrected whether miR-203 activates or inhibits the PI3K/AKT signaling pathway. We apologize for the inaccuracies here. The description here is now changed to ”In LPS-induced ALI mice, miR-203 can suppress the PI3K/AKT signaling pathway by inhibiting PIK3CA, a gene encoding a PI3K catalyst, to increase the downstream apoptotic proteins, and ultimately inhibit epithelial cell proliferation and promote apoptosis [97, 98]. And, repressed miR-203 effectively attenuated LPS-induced interstitial pneumonia.”
In addition, we have carefully checked the manuscript and revised some grammar, logic and other errors in the manuscript, please check. Thanks again for your constructive comments.
